# Molecular Prognostic Factors in Uterine Serous Carcinomas: A Systematic Review

**DOI:** 10.3390/curroncol32050251

**Published:** 2025-04-25

**Authors:** Anna Svarna, Michalis Liontos, Alkistis Papatheodoridi, Aristea-Maria Papanota, Eleni Zografos, Maria Kaparelou, Flora Zagouri, Meletios-Athanasios Dimopoulos

**Affiliations:** Department of Clinical Therapeutics, National and Kapodistrian University of Athens, Alexandra Hospital, V.Sofias 80, 11528 Athens, Greece; mliontos@gmail.com (M.L.); alkistispapath@gmail.com (A.P.); ampapanota@yahoo.gr (A.-M.P.); el_zogra@hotmail.com (E.Z.); mkaparelou@yahoo.com (M.K.); florazagouri@yahoo.co.uk (F.Z.); mdimop@med.uoa.gr (M.-A.D.)

**Keywords:** serous uterine cancer, prognostic factors, biomarkers

## Abstract

Uterine serous carcinomas are an aggressive minority of endometrial cancers. They are characterized by mutations in TP53 and extensive copy number alterations and are primarily classified in the copy number-high/p53abn molecular prognostic group, highlighting a unique molecular profile that is crucial for understanding their behavior and treatment responses. Clinical studies have shown that molecular categorization via biomarkers can facilitate proper treatment selection, and this is now widely used. In this context, the scope of this systematic review is to identify molecular characteristics with prognostic significance for these neoplasms to further inform on their treatment needs. We performed a comprehensive literature search of all articles written in English using the PubMed/Medline and Cochrane databases through February 2025. Our review led to the inclusion of 95 studies, from which we identified a total of 66 distinct molecular characteristics along with new cancer signatures that may impact prognosis. These findings have the potential to inform clinical practice by aiding in the development of tailored treatment strategies for patients with uterine serous carcinoma, ultimately improving outcomes in this challenging malignancy.

## 1. Introduction

Endometrial cancer is the most common gynecological malignancy in developed countries. In 2025, it is predicted that more than 69,000 women will be diagnosed with the disease in the United States alone. Uterine Serous Carcinomas (USC) constitute a subset of endometrial carcinomas, diagnosed in about 10% of cases. USCs are characterized by diagnosis at a more advanced stage, common recurrences outside the pelvis, and worse prognosis [1,2,3]. USCs, along with other high-grade histologies, were historically classified as type 2 endometrial carcinomas in distinction to low-grade endometrial carcinomas that were designated as type 1 lesions.

Nowadays, the classification of endometrial carcinomas has advanced through the comprehensive molecular analysis performed by The Cancer Genome Atlas (TCGA), recognizing four subtypes with specific molecular characteristics: POLE ultra-mutated, microsatellite instability hyper-mutated, copy-number low, and copy-number high [4]. In clinical practice, though, it is more common and accessible to test for the surrogate markers of these subgroups (e.g., p53 immunohistochemistry, microsatellite instability, and POLE proofreading mutation) that can serve as a practical approach to apply molecular classification. In the newest FIGO 2023 staging [5], molecular classification plays an undisputed role in stratifying patients into risk groups that inform patients’ need for therapy.

Following recent phase III trials, the addition of immunotherapy to the first-line chemotherapy has entered daily practice and is set to become the new standard of care for at least the dMMR subgroup of patients [6,7,8]. The use of molecular classification to stratify patients in order to better understand their clinical behavior has produced interesting and informative results. As expected, MMR-deficient (dMMR/MSI-high) patients yielded an astounding benefit from the addition of immunotherapy across all trials. However, when the p53mut subgroup of patients was examined, mostly comprising the serous subtype, the results were mixed [6,7,8]. That lack of understanding of USC and the need for further molecular classification of this patient group was thus underlined.

In this context, we undertook a systematic review of molecular characteristics associated with serous endometrial cancer that could provide insight into treatment improvement.

## 2. Materials and Methods

A bibliographic search in the PubMed/Medline and Cochrane databases was conducted using the keywords “(serous) AND (endometrial cancer OR uterine cancer) AND (prognostic factors)” according to the PRISMA guidelines [9]. A literature search produced 2331 results in total on 2 February 2025. Furthermore, in order to identify any additional eligible articles, reference lists were also meticulously examined, resulting in a total of 46 articles to be included, as shown in Figure 1. Duplicate articles were removed and two investigators (ML and AS) screened 2377 papers by reading the titles and the abstracts. In case of disagreement between the pair of investigators, team consensus was obtained after consulting the study coordinator (FZ). We selected 420 to read in full. Articles were considered eligible if they fulfilled the following criteria: 1. Written in English. 2. Included a molecular alteration that was associated with the patient’s prognosis directly or indirectly (linked with parameters that are considered prognostic factors). 3. Included in their statistical analysis a subgroup of USC patients or included a multivariate analysis about survival that took into consideration the histological type of the tumor. Of the 420 articles, 95 met those criteria and were included in this review. Factors that had more than one citation are analyzed in the text.

## 3. Results

The search strategy finally led to 95 eligible articles [10,11,12,13,14,15,16,17,18,19,20,21,22,23,24,25,26,27,28,29,30,31,32,33,34,35,36,37,38,39,40,41,42,43,44,45,46,47,48,49,50,51,52,53,54,55,56,57,58,59,60,61,62,63,64,65,66,67,68,69,70,71,72,73,74,75,76,77,78,79,80,81,82,83,84,85,86,87,88,89,90,91,92,93,94,95,96,97,98,99,100,101,102,103,104]. We found 66 molecular factors whose prognostic significance has been evaluated in USC patients. These molecular alterations could be grouped into seven categories according to their cellular function (DNA repair, membrane receptors, hormone receptors, adhesion molecules, cell cycle, cancer signatures, and immunogenicity) and are further analyzed below. The individual studies outlined in the text, categorized into seven groups—excluding cancer signatures—are presented in Table 1, Table 2, Table 3, Table 4, Table 5 and Table 6. An additional Appendix A includes additional potential prognostic factors not analyzed in the main text and can be found in the Appendix A. The results curated in the tables concern the entirety of the patient sample of the study unless noted as an independent predictor.

### 3.1. DNA Repair

#### 3.1.1. BRCA1/2

Mutations in BRCA1/2 genes have known predictive significance among patients with high-grade uterine serous ovarian carcinomas [105]. Several studies have evaluated the presence of BRCA1/2 mutations also in patients with USCs [106,107]. Although the analyses are less extensive than in ovarian cancer, BRCA1/2 mutations are recognized in approximately 5% of USC patients [106,108,109]. Other genes implicated in the Homologous Recombination Repair pathway are also infrequently mutated in USC patients (e.g., ATM 3.58%, PALB2 2%, CHEK2 0.76%) [109].

Based on the above, several studies have evaluated the predisposing, predictive and prognostic significance of BRCA1/2 mutation in USC patients. BRCA1/2 mutation carriers have a greater risk for developing breast and ovarian cancer, as part of the Hereditary Breast-Ovarian Cancer (HBOC) syndrome. Whether development of USC is part of the above hereditary syndrome has not yet been elucidated. Several studies have produced conflicted results [39,108,110,111]. These studies are, however, limited by the small number of evaluated patients [39,108,112] and incomplete molecular analysis for BRCA1/2 genes aberrations [110].

In addition, conflicting data exist regarding the predictive and prognostic role of BRCA1/2 mutations in USCs. Beirne et al., by using an appropriate scoring system, have shown that BRCA1 was statistically favorably correlated with PFS in univariate and in multivariate analysis that included age, stage and chemotherapy use [47]. This, though, was not confirmed in a different study that evaluated BRCA1 expression [44]. Finally, data for BRCA2 mutations are limited to a case report of a patient with stage IIIA disease that remained free of relapse for 18 months after primary surgery [107]. The prognostic significance of both BRCA1/2 mutations among USC patients has also been tested in several small studies; none of them were able to detect differences in the OS [39,69].

#### 3.1.2. HRR Status

As other genes implicated in the Homologous Recombination Repair (HRR) pathway are also infrequently mutated in USC, and learning from the important prognostic role that HRR status plays in ovarian serous carcinomas, a number of studies have tried to explore that landscape.

In a small series of 19 patients evaluating HRR status with OncoScan SNP array in USC, 10/19 samples were HRD. However, the HRD phenotype did not show a significant correlation with OS, although the number of patients was too small to draw safe conclusions [79]. In a different patient series with the same aim utilizing next-generation sequencing to detect mutations in HRR-related genes, 22 out of 60 USC patients were HRRmut with the most common pathogenic mutations on BRCA1/2 and ATM genes. It was shown that the HRRwt and HRRmut groups did not significantly differ in terms of PFS; however, HRRmut patients had a significantly better disease-specific survival (DSS) in late (St III-IV) stages [81].

Finally, as mentioned, USC is usually p53mut and categorized in the copy-number high stratification group which the authors stratified for p53. Both PFS and DSS were significantly longer for HRRmut patients, indicating that it can indeed be an independent prognostic factor in p53abn SEC [81], although more studies are needed to corroborate this result.

#### 3.1.3. TP53

ΤP53 is the most frequently mutated tumor suppressor gene in human cancers. Its prognostic value in gynecological malignancies has been thoroughly documented and many studies have tested its role in endometrial cancer in particular. In the serous histotype, more specifically, p53 is the most frequently detected molecular aberration, encountered in about 85% of cases [113].

Numerous studies have focused on the correlation between TP53 mutations, clinicopathological parameters indicative of prognosis, and survival end-points, namely OS, PFS, or DFS. Overwhelming evidence supports the notion that the TP53 mutation is an independent predictor of worse outcomes [10,16,18,22,27,49], and that is the reason that it is used as a stratification factor in the new guidelines and FIGO staging system. In addition, TP53 expression has been associated with negative expression of hormone receptors in the tumor, another predictor of poor survival in endometrial cancer [114,115], despite there also being contradictory data [65,116,117].

As previously noted, TP53 mutations have been linked with serous histology in as much as 85% of cases [113,115] and are thought to be part of the carcinogenesis process [18,118,119,120,121]. Therefore, the prognostic and predictive role of TP53 mutation/aberrant expression was shown in studies enrolling uterine carcinomas independent of specific histology [13,18,20,24,53,73,122].

However, even among patients with serous endometrial carcinoma it was found that overexpression of TP53 was significantly correlated with shorter survival [14,17,66]. There was also a trend for a shorter mDFS, PFS and recurrence of disease for patients overexpressing TP53 or its isoform γ [17,67], but that was not confirmed in later studies including only stage I and II patients with USC [23,45].

**Table 1 curroncol-32-00251-t001:** Studies exploring prognostic factors associated with DNA repair.

Prognostic Factor	Bibliography	Method	No. Patients/ No. USC Patients	Result
**P53**	Inoue et al. [10]	IHC	139/12	Five-year survival rate p53 overexpressed 60% vs. p53 not-overexpressed 87%
	Reinartz et al. [13]	IHC	128/11	Not an independent prognostic factor of survival
	King et al. [14]	IHC	22/22	Significantly shorter survival (*p* < 0.022)
	Hamel et al. [16]	IHC	221/8	Associated with compromised PFS (*p* < 0.001), independent prognostic factor for PFS
	Bancher-Todesca et al. [17]	IHC	23/23	Significantly shorter survival than those whose tumors did not (*p* = 0.033)
	Geisler et al. [18]	IHC	137/14	Independent prognostic factor of worse five-year survival in multivariate analysis (*p* = 0.0028)
	Salvesen et al. [20]	IHC	142/3	Independent prognostic impact (*p* < or =0.05)
	Coronado et al. [22]	IHC	114/27 non-endometrioid	p53 (*p* < 0.001) overexpression had a positive correlation with a high risk of recurrence, independent prognostic indicator of recurrence
	Lundgren et al. [24]	IHC	358/40 non-endometrioid	Significant predictor of relapse (*p* < 0.001), in multivariate analysis lost its prognostic capability
	Alkushi et al. [27]	IHC	200/13	Prognostic significance in the subset of patients with endometrioid carcinomas (*p* = 0.02), but not in patients with clear cell or papillary serous carcinomas
	Daniilidou et al. [48]	IHC	61/12	Not correlated with stage (*p* = 0.466)
	Vandenput et al. [43]	IHC	149/92 USC and clear cell	In metastatic or recurent patients correlated with survival (*p* = 0.01)
	Winder et al. [66]	IHC	313/313	Associated with worse overall survival (OS) HR, 4.20 [95% CI, 1.54–11.45]; p16: HR, 1.95 [95% CI, 1.01–3.75] and progression free survival (PFS) HR, 2.16 [95% CI, 1.09–4.27]
	Jia et al. [73]	IHC	212/77	p53 abnormalities correlated with worse disease-free survival (DFS) (*p* = 0.025). p53 (HR: 2.270, 95% CI: 1.124–4.586, *p* = 0.022) independently predicted DFS in non-EEC patients, not OS
**p53 isoform γ**	Bischof et al. [67]	quantitative Real-Time PCRs (RT-qPCR)	37/37	Relative p53γ expression to be associated with reduced PFS
**BRCA**	Bruchim et al. [39]	differential restriction	31/31	mOS 25 months, no significant differences in the mOS, two-year survival, or PFS between the mutation carriers and the noncarriers
	Kadan et al. [69]	DNA sequencing	64/64	mOS (25 vs. 37 months; *p* = 0.442), mPFS (37 vs. 29 months; *p* = 0.536), and mDSS (60 vs. 39 months; *p* = 0.316) were similar between the carrier and noncarrier groups
**BRCA1**	Amichay et al. [44]	IHC	52/52	Did not correlate to survival
	Beirne et al. [47]	IHC	72/72	Statistically significant decreased PFS when exhibiting tumor cell nuclei staining of 76% or greater (*p* = 0.0023)
**HRD**	Jönsson et al. [79]	OncoScan SNP array	19/19	No significant correlation with OS
	Dong et al. [81]	next-generation sequencing	60/60	Similar PFS (HR, 0.500; 95% CI, 0.203–1.232; *p* = 0.132), but significantly longer DSS in the tHRRmt patients than in the tHRRwt patients (HR, 0.176; 95% CI, 0.050–0.626; *p* = 0.007), when p53abn: both PFS and DSS were significantly longer in the tHRRmt (*p* = 0.040 and *p* = 0.008)

### 3.2. Membrane Receptors

#### 3.2.1. ErbB-2/HER2/Neu

ErbB-2 is a member of the epidermal growth factor receptor (EGFR) family of membrane receptors implicated in cellular proliferation, differentiation, and apoptosis [123]. Four members of the EGFR family are currently known and overexpression or structural alteration of these proteins have been found in a plethora of cancer types [123].

ErbB-2 gene amplification can lead to an up to 100-fold increase of HER2 protein expression in the cancerous cell, allowing for a ligand-independent activation of the receptor driving proliferation and cellular survival [124]. Since 1999, it was shown that HER2 amplification was associated with serous histological type [19]. Numerous studies have then evaluated HER2 gene amplification or protein overexpression in USCs with the percentage of the positive cases varying from 18 to 63%, dependent on the number of patients analyzed and the method used to assess HER2 status [19,29,31,32,62]. The concordance of the two methods (gene amplification and protein overexpression) has been shown to be lower than 50% of total HER2 abnormal cases.

In studies that included patients independent of uterine cancer histological types, both HER2 expression and amplification were significantly correlated with features of aggressive disease, such as higher stage, positive lymph node status, and greater than 50% myometrial invasion [32]. It seems, though, that HER2 gene amplification is also correlated with worse disease clinicopathological characteristics even among USCs [31,42,100], but that was not consistent in all studies, especially when immunohistochemistry was the single method used for the evaluation of HER2 overexpression [62].

Several lines of evidence support that among patients with localized USCs that receive radical surgical treatment, HER2 amplification is associated with increased risk of relapse [30,32,34,35,46]. More specifically, in a study of 169 stage I USC patients, HER2 positivity was significantly linked with the presence of LVSI, higher recurrence rates, and worse PFS and OS in both univariate and multivariate analysis [88]. In addition, HER2 overexpression/gene amplification is an adverse prognostic feature among patients with USC, as shown in the majority of the studies [15,28,32,35,37,42,46,62,75,100,125,126]. Also, a meta-analysis confirmed that HER2 overexpression is correlated with worse overall survival among endometrial carcinoma patients and retained its adverse prognostic significance for OS even among USC patients [58].

#### 3.2.2. EGFR

EGFR/ErbB-1 is also implicated in carcinogenesis in several carcinomas. In endometrial carcinoma, a significant correlation between EGFR expression and grade, metastasis, myometrial invasion, and age has been found [11]. Conflicting results exist regarding the prognostic significance of EGFR expression in both endometrial carcinomas in general and USCs in particular [11,12,13,37,127].

**Table 2 curroncol-32-00251-t002:** Studies exploring membrane receptors as prognostic factors.

Prognostic Factor	Bibliography	Method	No. Patients/No. USC Patients	Result
**EGFR**	Khalifa et al. [11]	IHC—overexpression	69/16	Significantly correlated with histologic grade (*p* < 0.001), metastasis (*p* < 0.001), cell type (*p* < 0.01), myometrial invasion (*p* < 0.01), independent prognostic factor when controlling for cell type
	Khalifa et al. [12]	IHC—positivity	69/16	Significantly correlated with nonendometrioid cell types and tumor metastases, survival from 86 to 27% (*p* < 0.03)
	Reinartz et al. [13]	IHC—positivity	128/11	Did not correlate with length of survival or known prognostic variables
	Konecny et al. [37]	IHC	279/134 non-endometrioid	Significantly associated with poor overall survival in patients with type II EC (EGFR, median survival 20 vs. 33 months, *p* = 0.028), retained prognostic independence when adjusting for histology, stage, grade, and age (*p* = 0.0197)
**HER2**	Khalifa et al. [12]	IHC—overexpression	69/16	Significantly associated with depth of myometrial invasion
	Saffari et al. [15]	gene amplification by FISH	92	Shorter overall survival than women whose endometrial cancer lacked amplification (*p* = 0.018)
		IHC—moderate/high expression	92	Lower cumulative overall survival by log rank analysis (*p* < 0.0001), independent predictor of overall survival (*p* = 0.0163)
	Hamel et al. [16]	IHC	221/8	Independent prognostic factor for worse PFS
	Rolitsky et al. [19]	gene amplification by FISH	72/7	Significant negative predictive value beyond stage, grade, and cell type (*p* = 0.002)
	Coronado et al. [22]	IHC	316	HER-2/neu (*p* = 0.018) overexpression had a positive correlation with a high risk of recurrence, not an independent prognostic indicator of recurrence
	Slomovitz et al. [28]	IHC	68/68	Overexpression associated with a poorer OS (*p* = 0.008)
	Santin et al. [29]	IHC	27/27	Short survival associated significantly with heavy HER2/neu expression (*p* = 0.02)
	Santin et al. [30]	gene amplification by FISH	30/30	Significantly shorter survival time from diagnosis to disease-related death
	Diaz-Montes et al. [31]	IHC HercepTest (DAKO)	25/25	Overexpression significantly associated with a worse survival outcome (HR = 6.58, 95%CI: 1.36–31.89, *p* = 0.02)
	Morrison et al. [32]	IHC	483/58	OS significantly shorter (*p* = 0.0001) in overexpression (median, 5.2 years) versus those that did not (median of all cases, 13 years)
		gene amplification by FISH	483/58	OS was significantly shorter (*p* = 0.0001) in amplification of HER-2 (median, 3.5 years) versus those that did not (median of all cases, 13 years)
	Odicino et al. [34]	IHC	12/12	Overexpression associated with a poorer OS and a very low relapse-free survival time
	Villella et al. [128]	IHC	26/26	Correlated with lower OS (*p* = 0.01)
	Singh et al. [34]	IHC	45/45	Did not reach statistical significance in OS and PFS, but had a hazard ratio (HR) of 1.5 in RFS
	Ren et al. [42]	IHC	36/36	Expression 2 + ~3 + significantly associated with advanced surgical stage and worse OS (*p* = 0.03 and *p* = 0.0023, resp.)
	Konecny et al. [37]	gene amplification by FISH	279/134 non-endometrioid	Not significantly associated with poor OS in patients with type II EC (HER2, median survival 18 vs. 29 months, *p* = 0.113)
	Togami et al. [46]	IHC	71/71	correlated with lower OS (*p* = 0.01), independent prognostic indicators for RFS (*p* = 0.022)
	Zhang et al. [58]	IHC	Meta-analysis	Correlated with worse outcome with a HR of 1.98 (95% CI, 1.49–2.62) for OS, and a HR of 2.26 (95% CI, 1.57–3.25) for PFS
	Chen et al. [62]	IHC	52/52	Not assosiated with prognosis
	Jamieson et al. [75]	sWGS, targeted panel sequencing, IHC	187/187 p53abn	Associated with worse outcomes
	Erickson et al. [88]	IHC	169/169	significantly more recurrences in the HER2-positive cohort (50.0% vs. 16.8%, *p* < 0.001), associated with worse PFS and OS (*p* < 0.001, *p* = 0.024), multivariate analysis: HER2 (+)associated with inferior PFS (aHR 3.50, 95%CI 1.84–6.67; *p* < 0.001) and OS (aHR 2.00, 95%CI 1.04–3.88; *p* = 0.039)
	Shao et al. [100]	IHC, FISH	77/77	amplification significantly associated with deep myometrial invasion (>1/2), and increased intra-epithelial and stromal density of CD20 + or CD8 + TIL (all *p* < 0.05), associated with poor OS and PFS only in univariate analysis

### 3.3. Hormone Receptors

Hormone receptors have been strongly associated with survival in endometrial carcinomas [129]. Despite this being considered a pathological characteristic of type I endometrial cancers, the expression of hormone receptors is quite frequent in USC as well [59]. Hormone receptors status, and specifically the expression of Progesterone Receptors (PR) and that of Estrogen Receptor isoform-a (ER-a), were found to be prognostic factors for a better RFS and OS [26,33,46,56,58,59,83]. However, there are studies that dispute those results and found no correlation with prognosis in such patients [45,64]. These results, however, should be reevaluated taking into consideration the recent molecular classification, as it is unclear if their association with USC patients’ survival will remain significant.

**Table 3 curroncol-32-00251-t003:** Studies exploring hormone receptors as prognostic factors.

Prognostic Factor	Bibliography	Method	No. Patients/No. USC Patients	Result
**Hormone Receptors**	Engelsen et al. [33]	IHC	200/100 non-endometrioid	Loss of hormone receptors significantly correlated with aggressive phenotype and poor prognosis
	Togami et al. [46]	IHC	71/71	Correlated with higher OS (*p* = 0.008), independent prognostic indicators for RFS (*p* = 0 *p* = 0.01), independent factor associated with OS (*p* = 0.044)
	Zhang et al. [58]	IHC	Meta-analysis	Pooled hazard ratios (HRs) of ER for OS, CSS, and PFS were 0.75 (95% CI, 0.68–0.83), 0.45 (95% CI, 0.33–0.62), and 0.66 (95% CI, 0.52–0.85). Combined HRs of PR for OS, CSS, and PFS reached 0.63 (95% CI, 0.56–0.71), 0.62 (95% CI, 0.42–0.93), and 0.45 (95% CI, 0.30–0.68)
	Przewoźny et al. [83]	IHC	103/15	Loss of ER and PgR expression connected with a poor prognosis.
**ERa**	Sho et al. [56]	IHC	33/33	Cancer-specific five-year survival rates without an expression 54.5% vs. with an expression 0.0% (*p* = 0.04); significant prognostic indicator in patients with USC (*p* < 0.05)
**ER**	Kobel et al. [59]	IHC	192/192	Not significantly associated with OS
	Karnezis et al. [64]	IHC	460/104	Associated with improved DSS
**PR**	Kobel et al. [59]	IHC	192/192	Significantly associated with favorable OS (log rank, *p* = 0.0024). PR expression was significantly associated with favorable OS independent of age, stage, center and lymph-vascular invasion in stage I and II USC (hazard ratio = 0.266, 95% CI 0.094–0.750, *p* = 0.0123)
	Karnezis et al. [64]	IHC	460/104	Assosiated with favourable outcomes [HR (CI) 0.39 (0.25–0.62) for DSS, *p* < 0.0001]

### 3.4. Adhesion Molecules

#### 3.4.1. L1CAM

L1CAM (L1 adhesion molecule) is a type I membrane glycoprotein of the immunoglobulin superfamily (IgSF) [130], which was found to be expressed in a plethora of human tumors, including endometrial cancers. L1CAM regulates cell adhesion and migration during the development of the nervous system. Thus, it is implicated in the adhesion and the migration of cancerous cells and linked to poor outcomes [131,132].

In TCGA analysis, high L1CAM was significantly correlated with advanced stage, grade, serous histology, positive cytology, positive pelvic and para-aortic lymph nodes, and shorter OS in univariate analysis of endometrial carcinomas [133]. In multivariate analysis, L1CAM retained its significance as an adverse prognostic factor for survival. Following studies confirmed that L1CAM expression is related to non-endometrioid and specifically serous histology, adverse clinicopathological features, and worse prognosis [61,64,131,134,135]. Interestingly, combined L1CAM and TP53 expression had a stronger prognostic value when considering distant recurrences [61].

#### 3.4.2. CD44

CD44 is a cell surface glycoprotein involved in several cellular functions, including cell proliferation, migration, metastasis, and immune response regulation. Due to its pluripotency and its relation to endometriosis, CD44 expression has been studied in endometrial carcinomas. However, neither an association with specific clinicopathological characteristics of the disease [25,136,137], nor a prognostic significance, has been shown so far [138]. Anti-CD44 targeted agents are currently in development, thus further evaluation of this protein in USCs is warranted.

**Table 4 curroncol-32-00251-t004:** Studies exploring adhesion molecules as prognostic factors.

Prognostic Factor	Bibliography	Method	No. Patients/No. USC Patients	Result
**CD44**	Hosford et al. [25]	IHC	32/32	No correlation to known prognostic features
**L1CAM**	Van Gool et al. [61]	IHC	116/30 non-endometrioid	Significant association with rate of distant metastasis (*p* = 0.018)
	Karnezis et al. [64]	IHC	460/104	Associated with poor outcomes (hazard ratio (HR) 3.35 [2.10–5.23] DSS, *p* < 0.0001)

### 3.5. Cell Cycle and Intracellular Signaling Pathways

#### 3.5.1. Cyclin D1

Cyclin D1 regulates the cell cycle and, more specifically, its progression through the G1 phase. Cyclin D1 has been implicated in the tumorigenesis of several neoplasms when overexpressed. Conflicting results exist regarding its role and prognostic significance in endometrial carcinomas [50,139,140,141]. Specifically for USCs, cyclin D1 expression is detected only in a minority of cases [142,143], and no prognostic information is available.

#### 3.5.2. P16

P16 retains a pivotal role in cell cycle regulation and functions as a cyclin-dependent kinase inhibitor. More specifically, P16 inhibits the cyclin D1-CDK4/6 complex from further promoting cell cycle progression through the G1 phase. In USPC patients, P16 loss is quite frequent [144] and has been correlated with non-favorable clinicopathological parameters [120] and worse PFS [21,66]. However, these data are based on a limited number of studies, and further investigation of the prognostic significance of P16 loss in USC patients should be conducted.

#### 3.5.3. Synuclein-γ

Synuclein-γ (SNCG) belongs to a family of proteins which is suggested to regulate cellular membrane stability. However, when examined in endometrial carcinomas, SNCG expression was significantly associated with tumor grade and stage, type 2 cancers, deep myometrial invasion, and lymphovascular invasion, and was linked with a shorter OS and DFS in univariate analysis [145]. SNCG is significantly more frequently expressed in USCs [38,146] and is also associated with shorter PFS, OS, and possible chemoresistance to paclitaxel [38,66]. However, SNCG expression was correlated in these tumors with that of TP53 and P16, and further studies are needed to clarify the relationship between these molecular pathways.

#### 3.5.4. PTEN-PI3K-AKT-FBXW7 Pathway

PTEN is a tumor suppressor gene that regulates cell cycle and proliferation. PTEN mutations are found in a plethora of cancers. In endometrial cancer, PTEN mutations are usually encountered in the endometrioid subtype, but in a series of USC patients, 36% harbored such a mutation [87]. Studies that evaluated PTEN immunoexpression in USC patients revealed no prognostic significance [48,147].

PIK3CA encodes the catalytic subunit of the PI3K that acts upstream of PTEN. In contrast to PTEN, PIK3CA amplification was significantly associated with serous histology [70]. In endometrial cancer, PIK3CA amplification was found to be associated with higher age, FIGO stage and grade, nodal metastasis and myometrial infiltration [70]. However, the results for the prognostic significance of either PIK3CA amplification or the presence of exon 9 or exon 20 missense mutations in patients with USCs are contradictory [55,60,62].

The PI3K-AKT pathway has been shown to be mutated in 30% of USC cases. In a study evaluating a cluster of biomarkers participating in the PI3K-AKT-FBXW7 pathway, 36 USC cancer patients were evaluated. When adjusted for stage, it was shown that higher expression of CHK1 (nuclear and cytoplasmic), FBXW7 (nuclear), and PPP2R1B (cytoplasmic) was significantly correlated with a decreased risk for progression [74]. Interestingly, in a different series of patients, the correlation of FBXW7—which interacts with PI3K pathway activation through mTOR stabilization—expression with the risk of progression, namely DFS, was not shown [62].

When PPP2R1A, which is implicated in the negative control of cell growth and division in the same pathway, is concerned in endometrial cancer, it has been correlated with poor survival and serous histology. However, the negative effect is more prominent in the endometrioid type rather than in serous carcinomas, where a clear correlation with survival has not been shown [62,87].

Due to the prevalence of the pathway mutation, new agents are being developed that target it, mainly in cell lines and xenografts with limited activity; however, this is an area of development.

#### 3.5.5. Tubulin-β-III

Tubulin-β-III has already been correlated with poor prognosis and overall survival in ovarian serous cancers, so the question of its value as a predictive marker in USPC patients is also valid. In endometrial cancer, however, the immunostaining pattern of tubulin did not correlate with age, clinical stage, histological grade, histology, depth of invasion, response rates, RFS or DFS in endometrial cancer [43,148]. The results were the same in the group of type II tumors as well. Lastly, in a study on USC patients after their treatment with Platinum/Taxane chemotherapy, the median OS among patients with high tubulin-b-III was significantly lower [51]. The question of its use as a predictive marker thus cannot be resolved as the evidence is insufficient.

**Table 5 curroncol-32-00251-t005:** Studies exploring prognostic factors associated with the cell cycle.

Prognostic Factor	Bibliography	Method	No. Patients/No. USC Patients	Result
**P21**	Salvesen et al. [20]	IHC	142/3	Influenced survival in univariate analyses (*p* < or =0.05), not independent prognostic impact
**P16**	Salvesen et al. [21]	IHC	316	Five-year survival of 47% for absent/minimal nuclear p16 expression vs. 81% for moderate/high nuclear p16 expression (*p* < 0.0001), independent prognostic factor
	Winder et al. [66]	IHC	313/313	associated with worse OS p16: HR, 1.95 [95% CI, 1.01–3.75] and PFS HR, 1.53 [95% CI, 0.87–2.69] compared with low levels
**Synuclein-gamma (SNCG)**	Morgan et al. [38]	IHC	20/20	Correlated with advanced stage and decreased PFS
	Winder et al. [66]	IHC	313/313	PFS rate at 5 years worse for high SNCG expression, at 40% vs. 56% for low SNCG expression (log-rank *p* = 0.0081; HR, 1.36; 95% CI, 0.96–1.92 in adjusted Cox model)
**class III beta-tubulin**	Vandenput et al. [43]	IHC	149/92 USC and clear cell	No correlation with recurrence or survival
	Roque et al. [51]	real-time PCR	48/48	Overexpression stratified patients by OS (copy number ≤ 400: 615 days; copy number > 400: 165 days, *p* = 0.049)
**PTEN**	Daniilidou et al. [48]	IHC	61/12	Not correlated with stage (*p* = 0.267)
	Karnezis et al. [64]	IHC	460/104	Not assosiated with outcomes
**CyclinD1**	Liang et al. [50]	IHC	201/21	High CyclinD1 expression associated with poor prognosis vs. patients without CyclinD1 staining (*p* < 0.05)
**PIK3CA**	McIntyre et al. [55]	Sanger sequencing of DNA	99/26	Not associated with shorter disease-specific survival (*p* = 0.57)
	Lemetre et al. [60]	RNA-sequencing	323/52	Not correlated with prognosis
	Holst et al. [70]	FISH	188	PIK3CA amplifications were associated with disease-specific mortality
	Jamieson et al. [75]	sWGS, targeted panel sequencing	187/187 p53abn	Associated with worse OS
**FBXW7**	Chen et al. [62]	IHC	52/52	Not assosiated with prognosis
	Dinoi et al. [74]	IHC	36/36	Associated with a decreased risk of progression, after adjusting for stage
**PPP2R1A**	Taskin et al. [65]	IHC	78/17	Significantly related to poor prognosis only in univariate analysis
	Hong et al. [87]	NGS	263/41	PPP2R1A mutations had significantly shorter survival than did those without mutations (*p* = 0.005 and *p* < 0.001)
**PPP2R1B**	Dinoi et al. [74]	IHC	36/36	Associated with a decreased risk of progression, after adjusting for stage

### 3.6. Cancer Signatures

The search for prognostic factors has led several researchers to try and find cancer signatures—a cluster of expressed or mutated genes—that can yield a prognostic role on serous endometrial cancer.

Chen et al. used USC RNA-seq data to find dysregulated genes in numerous cancer pathways. Using regression analysis, a four-gene signature (KRT23, CXCL1, SOX9, and ABCA10) was found to characterize a high-risk group with a significantly lower RFS and OS [76]. Interestingly, the four-gene signature’s ability to identify low-stage high-risk patients can offer important clinical insight and help tailor treatment.

Jamieson et al. used sWGS in p53mut endometrial cancers mostly comprising serous carcinomas. They identified five signatures that showed a correlation with prognosis [75]. Signature 5 was linked to BRCA1/2 copy number loss and exhibited characteristics akin to the HRD signature found in high-grade serous ovarian cancer (HGSOC). Signatures 3 and 4 were associated with increased ploidy levels and amplifications of CCNE1, ERBB2, and MYC, with PIK3CA mutations being more prevalent in signature 3. Meanwhile, signature 2 was found to be associated with endometrial patients exhibiting p53 mutations. Patients with signature 2 showed improved OS, while those with signatures 1 and 3 experienced poorer OS outcomes.

In that light, Tran et al. used TCGA RNAseq data to analyze and develop a 73-gene signature in USC patients that can predict prognosis. Indeed, their model USC73, along with the use of stage, yielded impressive results and could predict OS [90]. None of these “cancer signatures” are widely used in clinical practice; however, cancer biology complexity might indeed need a cluster of dysregulated genes to explain and inform on patient survival.

### 3.7. Immunogenicity

Immunotherapy (IO) has entered daily clinical practice as first-line therapy in combination with chemotherapy. In that light, factors adding sensitivity to IO are in the spotlight.

PDL1 is expressed in from 17% to 47% of USC; however, it is not clear that it is correlated with OS or RFS as results are contradictory [91,98]. A meta-analysis by Mamat et al. showed that PDL1 expression by tumor cells did not affect OS, whereas PDL1 expression in immune cells was indeed significantly associated with LVSI and a worse OS [80]. In a different series of high-risk EC patients, PDL1 was correlated with an improved RFS [91], results that were corroborated by a different patient series where PDL1 CPS (+)—not TPS (+)—were experiencing a significantly better OS [94]. Overall, it is evident that PDL1 expression as a prognostic factor has yielded inconsistent results [76,101,102,103,104], but it can possibly be used as a marker of response to immunotherapy as in other malignancies.

Furthermore, it was found that tumors with high levels of CD8-positive tumor-infiltrating lymphocytes (TILs) demonstrated a more favorable prognosis compared to those with low levels, even among tumors with abnormal p53 expression (trend for OS and significant difference for DFS) [82]. Moreover, CD40 expression, a TNF receptor family member playing a part in antitumor immunity, was shown to be correlated with worse OS (*p* < 0.05) in non-endometrioid histologies [98]. High levels of LAG-3 and TIGIT were associated with a better PFS and OS, with high TIGIT expression being an independent prognostic value for a better OS [95]. Other immune checkpoints that were evaluated (TIM-3, B7-H3, VISTA expression in immune or tumor cells, IDO1 in tumor cells) were not found to have a significant correlation with survival [94,95].

These findings underline these molecules’ intricate roles within the tumor microenvironment, which involves complicated interactions between intrinsic tumor characteristics and various cancer pathways. Their use as prognostic factors in USC might soon be undermined due to their possible use as markers of response to immunotherapy, forever changing the therapeutic landscape of the disease.

**Table 6 curroncol-32-00251-t006:** Studies exploring immune-related prognostic factors.

Prognostic Factor	Bibliography	Method	No. Patients/No. USC Patients	Result
**PDL1-PD1**	Mamat et al. [80]	Meta-analysis	516/84	Survival outcomes of PD-L1 high expression had a significant association with worse OS in immune cells (IC) but not in tumor cells
	Zong et al. [91]	IHC–TPS-CPS	833/113	PD-L1 TPS (+) but not in ICs or CPS, was associated with a favorable prognosis
	Chen et al. [94]	IHC–TPS-CPS	99/99	PD-L1 CPS (+) associated with improved OS (*p* = 0.038), no association between PD-L1 expression and survival was found using TPS
	Kucukgoz et al. [102]	IHC	53/17	Expression of PD-1 and PD-L1 expressions in tumor area associated with shorter survival (*p* = 0.006 and 0.001), PD-1 and PD-L1 expressions in microenvironment were not found to be related with survival
	Pasanen et al. [103]	Multiplex IHC	842/29	Advanced cancers showed more frequent Ca-PD-L1 positivity (*p* = 0.016), and CPS (*p* = 0.029) and IC ≥ 1% (*p* = 0.037) positivity compared with early disease
	Engerud et al. [104]	IHC	689/65	PD-L1 and PD-1 expression showed no impact on survival
**TIM-3**	Chen et al. [94]	IHC–TPS-CPS	99/99	No association with survival
**B7-H3**	Chen et al. [94]	IHC–TPS-CPS	99/99	No association with survival
**lymphocyte-activation gene 3 (LAG-3)**	Chen et al. [95]	IHC	94/94	High levels of LAG-3 expression associated with better PFS and OS than those with lower levels of expression (PFS, *p* = 0.03, OS, *p* = 0.04), multivariate analysis: high TIGIT expression had an independent prognostic value for better OS
**T-cell immunoglobulin and ITIM domain (TIGIT)**	Chen et al. [95]	IHC	94/94	High levels of TIGIT expression associated with better PFS and OS than those with lower levels of expression (PFS, *p* = 0.01, OS, *p* = 0.009)
**V-domain immunoglobulin (Ig) suppressor of T-cell activation (VISTA)**	Chen et al. [95]	IHC	94/94	No significant association with survival
**CD40**	Zhao et al. [98]	IHC	68/23	Correlated with worse OS (*p* < 0.05) in non-endometroid histologies

## 4. Discussion

This systematic review comprehensively evaluated molecular alterations associated with prognosis in USCs. TP53 mutations are the single most crucial genetic alteration found in these neoplasms. Data presented in this systematic review indicate that the TP53 mutation also has prognostic significance for USC patients [14,17,66], which is also evaluated from the perspective of the modern molecular taxonomy of uterine neoplasms categorizing p53mut tumors as high-risk. Also, translational analysis of the PORTEC-3 study highlighted the prognostic significance of TP53 mutations for uterine neoplasms [121]. Therefore, this biomarker will be merely used to diagnose USC patients, abolishing its prognostic significance for this subgroup. However, it is known that diverse functional consequences of TP53 mutations exist—loss of function and gain of function mutations [149]. In this context, further analysis of the type of TP53 mutations in USC is warranted.

Using the new FIGO molecular classification has changed daily practice. P53, the main mutation of USC, categorizes serous carcinomas in the high-risk group. However, there are a number of double classifiers that harbor either a POLE mutation or a dMMR phenotype along with a P53mut one. According to the classification and our data, these patients have a significantly better prognosis [57,150] and are categorized into the POLE or dMMR subgroup, accordingly changing treatment strategies.

Moreover, the sensitivity of these patients to immunotherapy can further optimize treatment and long-term survival. According to all immunotherapy studies that incorporated immunotherapy with first-line chemotherapy, dMMR patients experience the biggest survival benefit, with some being long-term responders even in the metastatic setting. Nevertheless, only a small percentage of USC are dMMR or POLEmut, and further prognostic factors are needed to delineate the subgroup of USC patients that will respond to IO. PDL1 expression can be the answer, according to our research; however, more studies need to prove its exact prognostic and predictive value in endometrial cancer.

This review also outlined the significance of three additional genetic alterations for the prognosis of USC patients: mutations in BRCA1/2 genes, Her-2 amplification andPIK3CA amplification/mutations. The level of evidence that supports the association of each of these alterations with USC pathogenesis and prognosis varies. However, it should be taken into consideration that these alterations are targetable and data regarding the use of analogous agents in USC patients are encouraging [151].

The exact role of BRCA1/2 mutations and HRD in general in USC pathogenesis remains disputed. Contemporary data indicate that only a fraction of USCs bear such genetic changes [106,108,109]. Given the existence of targeted agents (PARP inhibitors) that are approved for BRCA1/2 mutant patients in several cancers, including breast, ovarian, prostate, and pancreatic cancer, it would be interesting to delineate these genes’ role in USCs. Currently, PARP inhibitor olaparib is approved for metastatic patients as maintenance therapy in combination with Durvalumab following the DUO-E trial [152] regardless of BRCA mutation status. Interestingly, the BRCAmut patients in the study were almost equally divided between the dMMR and pMMR groups, perhaps highlighting a significant mutagenicity effect of the dMMR phenotype. This raises important questions about the potential benefits and efficacy of PARP inhibitors in USC patients who may harbor BRCA1/2 mutations. Further research is needed to explore the prevalence of these mutations in USC, their potential role as prognostic factors and their relationship with treatment outcomes when utilizing PARP inhibitors alone or in combination with immunotherapy.

By contrast, targeting HER2 amplification has already shown clinical benefit in USC patients with the addition of trastuzumab to first-line chemotherapy and with trastuzumab-deruxtecan as a later-line treatment option [151,153]. Analogously, with breast and gastric cancer, HER2 overexpression is a predictive factor for response to HER2-targeting agents [128,154,155] and platinum-based regimens [156]. More specifically, according to a randomized phase II study, the addition of the anti-HER2 antibody trastuzumab to the paclitaxel-carboplatin regimen increased both PFS and OS [151] among patients with advanced, recurrent, or metastatic USC with HER2 overexpression. Moreover, the recent results of the phase II Destiny-PanTumor02 trial entered the use of trastuzumab-deruxtecan in the treatment armamentarium in recurrent HER2 (+) endometrial cancer [153]. Finally, based on the potential active immune microenvironment of the HER2 (+) tumors and the known synergistic effect of anti-HER2 treatment and immunotherapy, such a combination in endometrial carcinoma may be a future development.

Importantly, this review summarizes data that HER2 amplification is not only predictive of anti-HER2 agents’ efficacy in USC patients but also prognostic for increased relapse rate and worse overall survival [58]. The latter prompts the evaluation of anti-HER2 treatments in patients with high-risk localized disease to benefit those patients further.

Furthermore, limited data exist in the literature regarding the prognostic role of PIK3CA amplification/mutations in USCs. However, there are several agents that target the PI3K/Akt/mTOR pathway and research towards this target is ongoing. In addition, PIK3CA amplification has been proposed as a mechanism of resistance to trastuzumab in Her-2 amplified USCs [157]. The above denote that the PI3K/Akt/mTOR pathway is a very attractive target in uterine cancer as well and many early phase clinical trials have been completed [158].

Finally, by utilizing new information techniques and applying them to the available mutational data of USC patients, researchers can develop cancer signatures by combining clusters of mutations to accurately predict patient survival. However, these results are preliminary and not yet widely available in clinical practice. Recognizing the complexity of cancer may provide a valuable everyday tool to further inform patients’ treatment strategies.

## 5. Conclusions

In conclusion, molecular alterations described in this review provide useful information regarding the prognosis of USC patients. Hopefully, many of these alterations could be used as predictive biomarkers for targeting agents in the near future. It is anticipated that this strategy will improve USC patients’ survival and quality of life. The heterogeneity of uterine carcinomas and even more that of uterine serous carcinomas is now clearly evident. However, we still need more clinical and translational studies in order to improve our understanding of the pathogenesis of the disease and, subsequently, its prognosis and treatment both at the local and extensive-recurrent stages.

## Figures and Tables

**Figure 1 curroncol-32-00251-f001:**
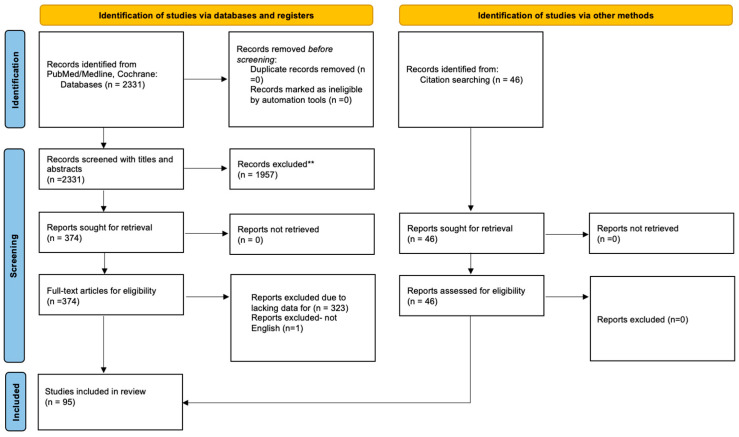
Search strategy depicted in a PRISMA flow diagram. ** did not meet inclusion criteria.

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
