# Peer review of "Molecular Prognostic Factors in Uterine Serous Carcinomas: A Systematic Review"

_curroncol, 2025, doi:10.3390/curroncol32050251_

Round 1

Reviewer 1 Report

Comments and Suggestions for Authors

See the added document

Author Response

We would like to thank the reviewer for his/her constructive comments.   Comment1: Tables 1 to 7 are difficult to read and should not be included in the text but put in an addendum. Also, there are several problems concerning these tables.
  • Some references refer to all type endometrial carcinomas and it is not clear if the data presented in the table refer to all type of endometrial carcinoma or only USC.
  • The authors should add a column with the number of cases evaluated.

Responce1: In the revised version of the manuscript, we have moved Tables adjacent to the relevant text in order not to overwhelm the reader, and table 7 has been moved to supplementary materials so as not to obstruct the flow of the text. We have added a column entitled “No. Patients/No. USC patients” in all tables so that the study sample of each paper is clear. Also, the phrase “The results curated in the tables concern the entirety of the patient sample of the study unless noted as an independent predictor.” has been added to the results area so that the tables content is better understood.

Comment2: In the results section, the different chapters are quite long and should be notably reduced (mainly the ones on BRCA1-2, HER2, HR, L1-CAM, Synuclein, PTEN, tubulin, immunogenicity…). Also, part of the results can be translated in the discussion section.

Responce2: Following the reviewer’s suggestion, we have reduced the length of the results section, and some passages (lines 406-420 and 423-433) have been moved to the discussion, as it was indeed more appropriate.

Comment3: Last, there are lot of minor points that needed to be reviewed:

  • Cochrane is misspelled throughout all text.
  • There are numerous words in the text that are cut: for example, “im-plicated” in line 108, “ex-pression” in line 171…
  • The sentence lines 46-47 is debatable and references should be added.
  • In line 109, the authors use the word infrequently. This word should be replaced by a percentage.

Responce3: Finally, all minor points mentioned by the reviewer have been reviewed and corrected.

Reviewer 2 Report

Comments and Suggestions for Authors

Extremely hard work for the authors to analyze and systematize a lot of information. But also for the readers, it is difficult to read and understand all data. I suggest to split the research in a series of 2 or 3 different articles.

Author Response

Thank you for your time and input.

Comment1: Extremely hard work for the authors to analyze and systematize a lot of information. But also for the readers, it is difficult to read and understand all data. I suggest to split the research in a series of 2 or 3 different articles.

Responce1: The aim of this review is to summarize all the potential prognostic factors regarding USC. As the paper is indeed long an effort was made to make it easier for the reader. The tables have been moved adjacent to the relevant text in order not to overwhelm the reader and table 7 has been moved to supplementary materials as not to obstruct the flow of the text. Moreover, the results section has been refined and some passages (lines 406-420 and 423-433)  have been moved to the discussion section to further support our conclusions.

Reviewer 3 Report

Comments and Suggestions for Authors

In general this is nice systemic review. Methodology is proper, references are relevant and well choosen. I would probably add some papers which are not directly in the focus of the analysis,however they would improve whole image of the problem, e.g. Uterine carcinosarcoma vs endometrial serous and clear cell carcinoma: A systematic review and meta-analysis of survival, by Raffone et al. DOI: 10.1002/ijgo.14033. But as I wrote this is not directly the matter of systemic review but to expand image. Moreover what would significantly improve value of the paper is presentation, what often seems to beproblematic part of review papers which cover many papers and the Readers are having problems with focus on the most important elements. But still those suggestions are to improve quite well written and organized paper which deserves publication. 

Author Response

Thank you for your time and input.

Comment1: Moreover what would significantly improve value of the paper is presentation, what often seems to beproblematic part of review papers which cover many papers and the Readers are having problems with focus on the most important elements.

Responce1:  We changed the layout of our paper, moving the tables adjacent to the relevant text so as not to overwhelm the reader. Table 7 has been moved to supplementary materials so as not to obstruct the flow of the text, which improves the presentation of our study.

Comment2:  I would probably add some papers which are not directly in the focus of the analysis,however they would improve whole image of the problem, e.g. Uterine carcinosarcoma vs endometrial serous and clear cell carcinoma: A systematic review and meta-analysis of survival, by Raffone et al. DOI: 10.1002/ijgo.14033. But as I wrote this is not directly the matter of systemic review but to expand image.

Responce2:  We evaluated your suggestion to add “Uterine carcinosarcoma vs endometrial serous and clear cell carcinoma: A systematic review and meta-analysis of survival, by Raffone et al. DOI: 10.1002/ijgo.14033.” and indeed presents the problem of the poor survival of the USC patients, however we believe it is outside the scope of this review and it was not added.

Round 2

Reviewer 1 Report

Comments and Suggestions for Authors

Thanks for the corrections.

The final version of the article is better and can be published.